# Innovative Application of Metabolomics on Bioactive Ingredients of Foods

**DOI:** 10.3390/foods11192974

**Published:** 2022-09-23

**Authors:** Sumei Hu, Caiyu Liu, Xinqi Liu

**Affiliations:** Beijing Advanced Innovation Center for Food Nutrition and Human Health, Beijing Engineering and Technology Research Center of Food Additives, National Soybean Processing Industry Technology Innovation Center, Beijing Technology and Business University, Beijing 100048, China

**Keywords:** food-derived bioactive ingredients, functional food, metabolomics, metabolites

## Abstract

Metabolomics, as a new omics technology, has been widely accepted by researchers and has shown great potential in the field of nutrition and health in recent years. This review briefly introduces the process of metabolomics analysis, including sample preparation and extraction, derivatization, separation and detection, and data processing. This paper focuses on the application of metabolomics in food-derived bioactive ingredients. For example, metabolomics techniques are used to analyze metabolites in food to find bioactive substances or new metabolites in food materials. Moreover, bioactive substances have been tested in vitro and in vivo, as well as in humans, to investigate the changes of metabolites and the underlying metabolic pathways, among which metabolomics is used to find potential biomarkers and targets. Metabolomics provides a new approach for the prevention and regulation of chronic diseases and the study of the underlying mechanisms. It also provides strong support for the development of functional food or drugs. Although metabolomics has some limitations such as low sensitivity, poor repeatability, and limited detection range, it is developing rapidly in general, and also in the field of nutrition and health. At the end of this paper, we put forward our own insights on the development prospects of metabolomics in the application of bioactive ingredients in food.

## 1. Introduction

The concept of metabolomics was proposed by Professor Jeremy Nicholsonrst for the first time in 1999, following proteomics, transcriptomics, genomics, and lipidomics [1,2]. It has become an important technology in the field of system biology since then [3]. Metabolomics mainly refers to the detection or identification of small molecules of cell metabolism at specific time or under specific conditions. These small molecules are collectively referred to as metabolites, including all the compounds produced or consumed by a metabolic process, such as amino acids, sugars, lipids, and organic acids [3,4]. There are two analysis methods, targeted metabolomics and non-targeted metabolomics [5]. Targeted metabolomics analysis is the identification and quantitative analysis of metabolites after selective extraction and purification, and the number of analyzed metabolites is relatively small [6]. Targeted metabolomics is generally used when it is necessary to determine changes in the content of one or more metabolites [7]. Non-targeted metabolomics analysis mainly focuses on the detection of more metabolites, which is conducive to the determination of new metabolites [6,8]. For example, untargeted metabolomics can be used to explore the possible presence of new metabolites [7].

Metabolomics has shown great development potential in food, medicine, nutrition, and other fields. It is widely used in the field of food, for the detection of metabolites in plants, cells or tissues, biological liquids, and other biological samples [9]. At present, metabolomics has been applied in food safety, food quality control, food traceability, food processing, food nutrition, health, and other areas [10,11,12]. Due to the large variety of chemical components of metabolites in food, current techniques can only detect a small amount of the metabolites, and the technologies for metabolomics analysis need to be further developed.

Techniques used for metabolomics analysis generally includes liquid chromatography-mass spectrometry (LC-MS), gas chromatography-mass spectrometry (GC-MS), and nuclear magnetic resonance spectroscopy (NMR) [13]. In recent years, the application of these technologies in food metabolomics is increasing, although there were drops since 2020 which was probably due to coronavirus (COVID-19) (Figure 1). These techniques have their own advantages and disadvantages for the metabolomics analysis of different substances. GC-MS and LC-MS are very sensitive and used more and more for metabolomics analysis [14]. MS is used to detect complex samples after separation and extraction. However, MS detection is selective to the polarity and volatile properties of the compounds and is destructive in the analysis of living samples. NMR is used to investigate metabolites at the atomic level, and it mainly includes hydrogen spectrum (^1^H NMR), phosphorus spectrum (^31^P NMR), and carbon spectrum (^13^C NMR) [13]. Heteronuclear NMR is widely studied, but it has many limitations compared to ^1^H NMR. For example, ^31^P NMR is mainly used for the measurement of phosphorus-containing metabolites, such as ATP, NADP, and GTP [15,16,17], and ^15^N NMR is mainly used for the measurement of nitrogenous metabolites, such as protein, DNA, and RNA [16,17], while ^13^C labeled NMR is mainly used for the determination of downstream metabolites in the study of cellular metabolites [18]. These isotope-enhanced NMR spectra can be used to assist in expanding the coverage of metabolites; however, the sensitivity is relatively low and the cost is really high [19]. Therefore, ^1^H NMR is the most widely NMR used in metabolomics at present, but the combination of heteronuclear NMR with other techniques may be a potential application direction. Metabolomics approach based on NMR spectroscopy is fast, simple, automatic and reproducible, and non-invasive, but its detection sensitivity is low, so a large sample size of >1 μM concentration should be guaranteed for the accuracy of the experiment [20,21]. At present, NMR is paid attention to in the identification of new metabolites and the analysis of metabolites flux in living cells [13]. NMR is still mainly used in the detection of metabolic components in living samples such as cells to protect samples and it is the best choice for real-time metabolic flux analysis [22]. In short, MS has been applied more and more in recent years because of its high sensitivity and its combination with chromatography, while NMR is still popular among researchers due to its overall efficiency and high throughput detection. In order to obtain realistic and accurate results, more and more researchers combine NMR with GC-MS and LC-MS to improve the accuracy of the studies.

This paper reviews the application of metabolomics in the area of food nutrition and health and reveals the health effects of food-derived bioactive ingredients. It provides a new perspective for the research and development of functional food and the treatment of chronic diseases.

## 2. Process of Metabolomics Analysis

There are a variety of compounds in the metabolome, but there is still no good technology which can identify the components of these compounds fully and effectively at present. Metabolomics, as an important part of life science and biological system, has made significant contributions to the bioactive ingredients in food and their health effects on human beings through the analysis of changes in the metabolites [10]. Metabolomics is used to separate and identify small molecules in blood, urine, feces, cells, culture media, or food ingredients, and to study the related pathways [23]. The molecular weights of these small molecules are generally under 1000 Da [3].

The procedure of metabolomics analysis mainly involves the following steps: sample preparation, metabolite extraction, derivatization, separation and detection, and data processing (as shown in Figure 2) [3]. Small changes in any of these steps may have a significant impact on the final results.

### 2.1. Sample Preparation

Sample preparation is the first step in metabolomics study. The quality of the prepared samples is the key to the success of metabolomics analysis, so it is very important to choose the appropriate preparation method for different samples.

The preparation methods are different for samples of various sources [24]. Keeping metabolite compositions of the original samples unchanged as much as possible and finding a suitable detection technology platform are two main problems in sample preparation. After the pretreatment of samples, choosing a suitable detection method can make the detection results have better repeatability and extraction efficiency [25]. For example, 4-chlorophenylalanine can be used to normalize the sample before GC-MS based metabolomics treatment to improve the extraction accuracy and efficiency [25]. Before the LC-MS based metabolomics sample is processed, the metabolites can be divided into different components by the mixed mode solid phase extraction method, and the appropriate column is selected to analyze the sample to improve the detection range of metabolites [26]. Methanol extraction can improve the quality of NMR spectra [27]. With the optimization of pretreatment technology, some special sample preparation methods have also been proposed. Solid phase microextraction (SPME) is widely accepted as a non-destructive method for the preparation of liquid samples [28]. Tijana Vasiljevic and colleagues proposed a method for preparing small samples of miniaturized SPME tips, which are coated with HLB particles [29]. It was the first study to analyze caviar samples using small SPME and LC successfully, and there was good extraction efficiency [29]. Wan Chan and colleagues compared the performance of several different serum preparation methods based on UPLC-MS, and found that serum samples prepared with methanol generated more accurate data [30]. In another study, it is showed that the speed of ultra-centrifugal treatment had a significant impact on the metabolic profile of fecal water; in particular, the concentration of P-Cresol changed with the increase of rotational speed [31]. However, this method is only suitable for NMR metabolomics studies at present.

For solid samples, freeze drying and grinding are required to reduce moisture in the samples and increase the release of metabolites, respectively. Quenching is a very important step to stop the metabolic processes, and this step includes adding liquid nitrogen, freezing, heating, and adding acid [32]. The omission of this step may cause changes in the metabolite composition by residual enzymes. However, time control is necessary in this step [32,33]. Sample preparation is a key step in metabolomics analysis [34]. How to prepare samples quickly without changing the original metabolite composition and make the operation repeatable are the problems to be solved in the future.

### 2.2. Metabolite Extraction

The step of metabolite extraction is generally the most rate-limiting step in metabolomics analysis [35]. There are different extraction methods for different types of samples to maximize the number, type, and concentration of the target metabolites. The selection of extraction solvent also has a significant effect on the recovery rate and metabolic profiles. The extraction solvents commonly used include water, chloroform, perchloric acid, methanol, acetonitrile, and other solutions [36,37]. It is necessary to choose hydrophilic solvents such as water-alcohol solutions for polar metabolites and hydrophobic solvents for non-polar metabolites. Estelle Martineau et al. compared the extraction efficiency of methanol/CHCl_3_/H_2_O, Acetonitrile/H_2_O, methanol/H_2_O, and Perchloric acid on mammalian cell metabolites, and found that using methanol/CHCl_3_/H_2_O for extraction can extract more metabolites, with good repeatability [35]. Karsten Seeger proposed a new method which extracts metabolites directly from NMR tubes by slice selection after centrifugation, and it provided a new idea for rapid determination of metabolites [38]. The most important thing of this method is that it could extract as many stable target metabolites as possible without adversely affecting subsequent analytical experiments [38]. 

### 2.3. Derivatization

This step is not always necessary. Generally, derivatization of the metabolites is required to transform the non-volatile compounds into volatile compounds to facilitate the analysis of metabolites and improve detection ability of the metabolites effectively if using GC-MS [39]. For example, the physicochemical properties of compounds with low ionization rate were changed by chemical derivatization to improve their ionization rate. Sezin Erarpat and colleagues used ultrasonic-assisted ethyl chloroformate to derivate l-methionine extract in human plasma and the recovery rate was up to 97.8 to 100.5% using GC-MS which could be regarded as a green and economical method [40]. Stable isotope labeling derivatization (SILD) is a novel sample pretreatment technology proposed in recent years, with a great potential in food metabolomics research based on LC-MS [41]. Shuyun Zhu et al. investigated a derivative method based on quadruplex stable isotope and developed 3-N-(D_0_-/D_3_-methyl-, and D_0_-/D_5_-ethyl-)-2’-carboxyl chloride rhodamine 6 G derivatization reagent, which can quickly and accurately quantify panaxadiol and panaxatriol in food [42]. Several studies have shown that derivatization can improve the ability of metabolite detection [43,44].

### 2.4. Separation and Detection

Separation and detection are important steps in metabolomics analysis. In the field of food nutrition, common separation technologies mainly include GC, LC, and capillary electrophoresis [13]. The separation of compounds is based on the adsorption capacity of each molecule in the stationary phase, and it is also related to the selection of column, eluent, fixation, and flow equivalence parameters [45]. In order to separate more metabolites, it is necessary to choose appropriate separation modes according to the polarity of the compounds. The separation technology is usually combined with high throughput detection technology to obtain large amounts of data. The commonly used detection techniques are NMR and MS [13]. Although the sensitivity of NMR is low, it can be used for non-invasive, rapid, and repeated analysis of a variety of metabolites at the μM levels [46]. It is simple to operate and suitable for high-throughput untargeted metabolomics analysis [47]. Both primary metabolites including amino acids, sugars, lipids, and organic acids, and secondary metabolites including flavonoids and alkaloids can be detected by NMR. By contrast, the sensitivity of MS is much higher, and it requires only a few μL of samples for analysis. MS can be combined with different separation techniques or in series according to different sample types [13]. GC-MS is mainly used to identify volatile and semi-volatile metabolites, while substances without volatile properties need to be derivatized, separated before detection by GC-MS. However, GC-MS cannot recognize any secondary metabolites [34]. Unlike GC-MS, LC-MS does not require complex pretreatment of samples, and it can directly separate and detect metabolites after extraction [34]. LC-MS is more comprehensive in metabolite identification and can determine secondary metabolites such as flavonoids as well as primary metabolites such as amino acids in plants.

Although current metabolomics techniques generally use a single detection tool, each technique has its own advantages and disadvantages. In order to identify and characterize more metabolites, combination of NMR and MS may achieve greater results. Manuja Kaluarachchi and colleagues identified metabolites in human plasma and serum by combination of ^1^H NMR and UPLC-MS [14]. They identified 4 metabolites with significant differences in plasma and serum by ^1^D NMR, and 10 other significant different metabolites by UPLC-MS, and most of them are found on glycerophospholipids [14]. Dong-sheng Zhao et al. determined the mechanisms of dioscorea bulbifera rhizome (DBR) on rat hepatotoxicity by integrating GC-MS and ^1^H NMR, and obtained a new potential therapeutic target, thus achieving an effective application of multi-platform metabolomics technology [48]. In addition, the introduction of chemicals in NMR tubes increased the likelihood of identifying compounds with specific physical and chemical properties; the ^15^N-edited NMR enabled specific binding to compounds containing free carbonyl [49]. The method of metabolic fingerprint analysis based on ultra-high-performance liquid chromatography–high-resolution mass spectrometry (UHPLC-HRMS) was optimized by using ethylene bridged hybrid C_18_ column, which showed good chromatographic resolution and realized the effective detection of infected metabolites in wheat [50]. Moreover, the optimization of parameters has been gradually studied. The researchers compared the Isotopologue Parameters Optimization (IPO) processing and manual processing of the original HPLC-TOF-MS data, and the parameters selected by IPO showed higher repeatability, and therefore it can be used to evaluate the optimum XCMS [51]. However, IPO need to take several days or even weeks to calculate the optimization parameters. In contrast, AutoTuner gives more robust and high-fidelity results [52]. MetaboAnalystR 3.0 is proposed as a new optimization process, which can not only optimize and correct parameters effectively, but also predict active pathways accurately [53]. In recent years, a hybrid metabolomics method based on mass spectrometry also attracted much attention. By bridging the advantages of targeted and untargeted metabolomics, more accurate results and more metabolites can be gained [54].

Different analytical instruments have different emphases. Considering the characteristics of samples and different analytical methods, a variety of separation and detection instruments can be used together to make the obtained metabolic data more comprehensive.

### 2.5. Data Processing

Data processing is an essential step in the process of metabolite screening, through which the changes of metabolites can be visualized and the possible metabolic pathways leading to these changes can be investigated using the KEGG database. Statistical analysis can help us to understand the metabolites in food and their impact on human health. Identification of metabolites is the most challenging step in metabolomics analysis [55]. The metabolites in the samples were obtained by comparing with the data in various resource databases. Choosing the right data processing method can greatly improve the accuracy of data analysis. There are many metabolome databases such as Metlin [14], Human Metabolome Database (HMDB) [56], KNApSack Database [57], and MassBank [58]. After aligning the information with these reliable databases and with multivariate statistical analysis, the obtained raw data can be converted to more meaningful conclusions, such as biomarkers.

Multivariate statistical analysis methods include principal component analysis (PCA), partial least squares discriminant analysis (PLS-DA), orthogonal partial least squares discriminant analysis (OPLS-DA), least absolute shrinkage and selection operator (LASSO), linear discriminant analysis (LDA), and so on [59,60]. Among which, PCA and PLS-DA are the most commonly used statistical methods in the field of metabolomics. PCA is a commonly used unsupervised dimensionality reduction method for metabolite quantity analysis, which reduces the data set to fewer dimensions to obtain greater variance [61]. It can help us to visualize the metabolic data, trend, and cluster. It has been reported recently that PCA was used in combination with quadrangular discriminating analysis (PCA-QDA) to identify the MS data of cancer samples, and its accuracy and specificity reached more than 90%, and therefore it can be called a satisfactory classification model [62]. PLS-DA is a supervised statistical analysis method that maximizes the correlation between variables, and it is often used to screen metabolites and to analyze overall metabolic changes between groups [63]. The availability of PLS-DA is good, and it can be used to process multiple dependent categorical variables simultaneously. However, PLS-DA is prone to overfitting [63]. In order to avoid this problem, based on the advantages of PLS-DA, OPLS-DA can divide the data into Y-related variation and Y-independent orthogonal variation and eliminate the variables unrelated to the experiment [64]. R^2^ and Q^2^ parameters are used to evaluate the prediction ability of the OPLS-DA model, and variable importance in the projection (VIP) can be generated from the model. VIP > 1.0 indicates that there are important potential biomarkers in the OPLS-DA model [65]. The authors compared PLS-DA with OPLS-DA in terms of model fitness and interpretability; although both are applicable, OPLS-DA had a higher interpretability [66]. At present, the co-analysis of PCA and OPLS-DA has become the mainstream trend of metabolomics to discriminate samples. Combination of more analytical models may be a future direction. Currently, the application of OPLS-DA is mainly to screen and identify biomarkers through s-plot/s-line, permutation, and VIP [67,68]. Using these methods to investigate the changes of metabolites may be a development direction in the future.

LASSO is a model selection method, and it can predict the phenotype by regression analysis of metabolites [69]. LDA can classify the samples according to the source and maximize the linear separation of the classes [70]. Kaitlyn M Mazzilli and colleagues evaluated the effects of various daily diets intake on serum metabolism using LASSO and found 102 related metabolites [71]. Virgilio Gavicho Uarrota et al. identified the metabolic components of cassava postharvest physiological deterioration (PPD) through PCA and PLS-DA models and realized good sample prediction [60]. The results provided good evidence for the metabolic differentiation of cassava during PPD. Moreover LDA and PCA in cluster analysis were considered to be suitable methods for distinguishing sex differences from organ differences [72]. For example, argininosuccinate showed significant differences between males and females in kidney tissue, and in the ventricle, males had significantly higher levels of free carnitine and total esterified carnitines than females [72]. So, targeted metabolomics is a good technique to test sex differences.

## 3. Application of Metabolomics in Nutrition and Health

Healthy diet has received widespread attention nowadays. People are gradually aware of the nutritional role of some food-derived bioactive ingredients in the prevention and regulation of chronic diseases [73]. Bioactive ingredients from plants and animals have been used in the development of functional foods and the treatment of diseases, such as peptides, polyphenols, and lipids, which are widely found in food and medicine. César G. Fraga et al. reviewed the health effects of polyphenols on diabetes and cardiovascular diseases, and their interactions with other bioactive components, and showed that eating enough polyphenol-rich foods can regulate chronic diseases and bring health benefits to humans [74]. Subhadeep Chakrabart et al. introduced the food sources of various bioactive peptides, including milk, eggs, soybeans, wheat, and fish, which can be used as lead compounds in the development of health supplements and functional foods [75]. However, the healthy nutritional effects and bioavailability of bioactives in food need to be characterized by some parameters [73]. Metabolomics, as a new omics technique, plays an important role in the qualitative and quantitative analysis of metabolites [76]. Here, we introduce the characterization of bioactive ingredients in foods by using metabolomics techniques and the health effects of these bioactive ingredients on cells, animals, and humans. Metabolomics has made an important contribution to the development and utilization of functional food. The applications of metabolomics in food-derived bioactive ingredients are summarized in Table 1.

### 3.1. Application of Metabolomics in the Discovery of Bioactive Substances in Plants

People concern about nutrition and high quality nowadays. Application of metabolomics in foods can help us to understand the biochemical indicators in food, facilitate the production and development of healthy food, and alleviate the problems affecting human health due to nutrition [105]. Here, we mainly introduce the bioactive substances in plants. Metabolomics characterizes the relationship between genotype and phenotype, and it has a good predictability [106].

Plants store nutrients differently at different growth stages. For example, many biochemical indexes change before and after seed germination, and the accumulation of metabolites is different [107]. Moreover, studies have shown that bioactive chemicals were higher in young leaves and epidermis parts of plants than in the mature parts [108,109].

Metabolomics can be used not only to screen valuable bioactive substances, but also to analyze the metabolite changes in different maturation stages and other processes. Studies have shown that some micronutrients, such as isoflavones and polyphenols, were significantly increased after soybean germination or sprout, with increased antioxidant properties [110]. A previous study analyzed the metabolite changes of two soybean seeds before and after germination by LC–ESI–MS/MS and found that the main different metabolites were flavonoids [77]. A total of 114 flavonoids metabolites were found in two soybean seeds and sprouts, including isofavones, dihydrofavone, chalcones, tannin, flavonoid carbonoside, flavonoid, anthocyanins, and other flavonoids [77]. With the information of the changes in the metabolites, they were able to detect six main metabolic pathways and generate more valuable information for the area [77]. Xiangyu Wu et al. investigated metabolite changes of mung bean at different germination stages using NMR and identified 63 metabolites in seeds and culture medium [78]. There were significant differences in amino acids, sugars, choline, and some secondary metabolites at different stages of germination [78]. Chunhui Xu and colleagues analyzed the relationship between the changes of total phenolic content (TPC) and total flavonoid content (TFC) and their antioxidant activities in vitro, as well as the inhibitory effects of α -glucosidase and pancreatic lipase in the process of green tea bud ripening, using UPLC-QTOF-MS [79]. PCA and hierarchical clustering analysis (HCA) showed that there were significant differences in metabolites among green tea samples with different ripening degrees, while PLS-DA and OPLS analysis showed that the levels of flavonoids and polyphenols decreased with the maturation of tea plants [79].

Another study found 35 metabolites total in the extract of *Lupinus albus* fractions by ^1^H NMR from four different fractions [80]. The 2,2-diphenyl-1-picrylhydrazyl radical (DPPH) scavenging activity and α-glucosidase inhibition activity of chloroform fractions were significantly higher than those of methanol, ethyl acetate, and hexane fractions [80]. Then, UHPLC-ESI-MS/MS was used to analyze and evaluate the organic extract, and 21 different metabolites were identified, among which isoflavones and alkaloids were the main components [80]. Through metabolomics, effective extraction fractions can be found to improve the utilization of bioactive components. Yan Cui et al. applied targeted metabolomics to screen components from hawthorn metabolites which inhibit thrombin activity, using UHPLC-Q-TOF/MS and LC-MS/MS, and discovered seven bioactive markers which were then investigated for their inhibitory effects on thrombin and possible key active sites [81]. With targeted metabolomics, researchers were able to find functional related bioactive substances.

The digestion process also has an effect on metabolites. Merve Tomas and colleagues conducted metabolomics analysis on the components of four cruciferous micro-vegetables before and after simulated gastrointestinal digestion in vitro by UHPLC-QTOF [82]. Compared with fresh samples, even after simulated digestion in vitro, the concentrations of total polyphenols, glucosinolates, and total monomeric anthocyanins in the four micro-vegetables decreased [82]. However, it proved that these four micro-vegetables contain bioactive substances that are beneficial for human health [82]. Similarly, the metabolic profile of red beet and amaranth after simulated gastrointestinal digestion during different storage periods were measured using UHPLC-QTOF [83]. The total polyphenol content and total antioxidant capacity of both microgreens also decreased after gastrointestinal digestion [83]. The results of the above metabolomics studies indicated that the practical application value of these bioactive substances is affected by gastrointestinal digestion. However, the reasons for this change may require further research.

These studies showed the application of metabolomics in the changes of bioactive substances in plants during different processes. These studies are helpful to the development of new nutritional foods. However, there are still many problems. For example, the enrichment method of the bioactive substances, the consumption ranges, the possible side effects, all need to be further explored. Although the metabolites and phenome of plant and animal food ingredients have been well reported, the role of these metabolites in the human body remains to be determined. The changes of these nutrients after digestion in human gastrointestinal tract and the extent to which they play a role in the human body, as well as the impact of individual dietary patterns or habits and other related metabolite changes need to be further studied. The functions of bioactive ingredients also need to be further purified and investigated using metabolomics, which will be a huge challenge.

### 3.2. Application of Metabolomics in the Effect of Bioactives In Vitro

Metabolomics was mainly applied in vivo at the beginning, and it has been gradually used in in vitro studies [111]. For the determination of different bioactive substances, especially substances far from the current human normal diet, cell experiments can help us to investigate the effects of these substances on human physiological conditions [112]. Bioactive substances are taken up by the cells and undergo various transformation reactions in the cells and their changes within the cells reflect the effects of the compounds. The effects of bioactive substances in some potential food ingredients on cellular metabolism have been reported previously [113,114].

Abhishek J. Gupta and colleagues analyzed the effect of nutritional components of soybean protein hydrolysate on the Chinese hamster ovary (CHO) cells growth and total immunoglobulin (IgG) production and identified 410 metabolites from soybean hydrolysates by UHPLC/MS/MS_2_ and GC/MS [84]. Jason Richardson et al. analyzed the different markers in soybean hydrolysates used in cell culture and their changing mechanisms through seven different LC-MS/MS identification methods and detected a total of 125 metabolites and 4131 short peptides of different lengths [85]. By comparing the growth of two different CHO cell lines, untargeted metabolomics was used to investigate markers of different batches of soybean hydrolysates and analyze the performance of soybean hydrolysates [85]. In another study, Jia-hui Nie et al. compared the effects of chemical components in the coat (RKBC) and kernel (RKBK) of red kidney bean extracts on the proliferation of B16-F10 cells, and found that there were significant differences in the content of component among organic acid/amino acid region, sugar region, and aromatic region, using NMR [86]. Furthermore, RKBC contains a variety of phenolic substances, which inhibited the proliferation of B16-F10 cells. Application of metabolomics contribute significantly to the development of functional food and drugs to inhibit melanoma [86].

The effects of different concentrations of bioactive substances on cells can also be analyzed by metabolomics. Different metabolites in cells treated with different concentrations of Chrysophanol-8-o-β-D-glucoside (CG) were found significantly separated by UPLC-MS/MS, and 42 major differential metabolites were selected [87]. Through multivariate statistical analysis, 26 of the 42 major metabolites were identified as differential metabolites between the high concentration group and control group, and 23 of them were differential metabolites between the low concentration group and medium concentration group. These differences were mainly due to differences in amino acid metabolism pathways. Another study identified a total of 30 metabolites from the control, positive, and exendin-4 treatment groups by NMR, of which 11 were differential metabolites [88]. Further pathway analysis revealed that the damaging effect of tert-butyl hydroperoxide (t-BHP) on cells was mainly manifested by the changes in intracellular glutamate, lactate, and energy production, while exendin-4 ameliorated these changes and protected the cells. Hui Li and colleagues investigated the effects of semi-inhibitory concentration of vitamin C on metabolism of mouse macrophage RAW264.7 and human myeloid leukemia cell line K562 [89]. The results showed 25 differential metabolites in RAW264.7 cells and 6 differential metabolites in K562 cells [89]. With the changes of these metabolites, they determined the safety and efficacy of high concentrations of vitamin C [89]. 

Chen et al. explored the toxic effect of emodin on HepG2 cells by using non-targeted metabolomics method based on NMR [90]. After treatment with different concentrations of emodin, they found 33 metabolites in cell extracts and 23 metabolites in cell medium [90]. They showed that 100 μM emodin had a significant inhibitory effect on HepG2 cells, and these effects involved eight related metabolic pathways [90]. Similarly, Matthieu Dallons and colleagues also found that 0.3 μM doxorubicin caused the separation of metabolites, and 23 metabolites were detected by ^1^H NMR [91]. With further study, the authors identified mitochondrial metabolic dysfunction was an important factor in cardiotoxicity caused by dexrazoxane [91]. All the above studies showed that metabolomics can be used to determine the effects of different concentrations of bioactive substances on cells, which can be regarded as a new method for drug evaluation or drug toxicity targets screening. 

Metabolomics is an important technique in the investigation of the regulatory effects of the food-derived bioactive ingredients. More evidence is still needed to understand the effects of these bioactives in humans. For example, in addition to the analysis of the composition of the culture medium, metabolomics can also be used to analyze the composition of the supernatant or lysate of the cells. Moreover, the utilization of food-derived bioactive ingredients after the transformation of intracellular substances needs further study. Future studies also need to validate the effects of different bioactives in vivo.

### 3.3. Application of Metabolomics in the Effect of Bioactives in Animals

Metabolomics plays an important role in understanding the physiological and pathological states of animals and human beings and discovering specific biomarkers [80]. Urine is the most commonly used body fluid for metabolomics analysis because urine contains less protein, and it does not need complex preparation. Serum, plasma, and tissue homogenate are also used for metabolite measurements [115]. The effects of bioactive substances from some potential food ingredients in animals have been reported using metabolomics approach previously [116,117].

Quinoa, as a crop with high nutritional value, is gradually gaining popularity. Although various health effects of quinoa saponins have been reported, its safety is still controversial. Ruoyu Zhang and colleagues investigated the changes of metabolites in urine of rats of different genders under the treatment with saponins of chenopodium quinoa wild, using UHPLC–MS [92]. They found that the same dose of quinoa saponins produced different toxic effects in rats of different sexes [92]. However, in the non-toxic range, quinoa saponins have certain application value. The study also suggested that metabolomics can play a role in gender-specific studies.

Corn silk has been reported to be effective in the treatment of diabetes [118]. The effects of corn silk on serum metabolites of diabetic rats were investigated using UPLC-ESI-Q-TOF/MS, and the results showed that 26 metabolites altered, with the levels of phosphatidylcholine and palmitic acid, the markers of diabetes, decreased [93]. Fenugreek flavonoids play a role in the regulation of diabetes, and 19 different flavonoids were identified in fenugreek using UPLC-Q-TOF-MS, of which 11 were metabolic biomarkers, mainly involved in lipid metabolism and amino acid metabolism, and the hypoglycemic function may be related to its protective effect on kidney [94]. Caijuan Zhang et al. examined the changes of metabolites in urine of diabetic mice after resistant starch 3 (RS3) and metformin treatment using UHPLC-LTQ/Orbitrap MS and identified 29 biomarkers which were involved in amino acid metabolism, lipid metabolism, and other pathways [95]. Fei Ren and colleagues analyzed the changes of serum metabolites of ganoderma amboinense polysaccharide (GAP) in mice with non-alcoholic fatty liver disease induced by high fat diet by LC-MS [96]. There were 18 biomarkers identified by hierarchical clustering and these metabolites involved in amino acid metabolism, TCA cycle, lipid metabolism, fatty acid metabolism and synthesis, and so on [96]. According to the results of metabolomic analysis, it is possible that GAP may protect the mitochondrial metabolic function [96].

Echinacea (L.) Moench (EP) has a control effect on the tumor growth of hepatoma mice [119]. The serum metabolic profile of mice after EP treatment was investigated by LC-MS and the results showed that the 12 differential metabolites were significantly improved, mainly involving in the biosynthesis of phenylalanine, tyrosine, and tryptophan and the metabolism of phenylalanine [97]. The effective bioactive ingredients can be obtained by combining molecular docking [97]. The hydrolysates of yak bone glue play a role in the regulation of obesity. The effects of the hydrolysates of yak bone glue on fecal metabolites of high-fat obese mice were studied by UPLC-QTF/MS, and 666 and 705 metabolites were detected in the positive and negative ion modes, respectively [98]. Further analysis showed that the anti-obesity effect of the hydrolysates of yak bone glue may be related to the changes of intestinal microorganisms, mainly involving in the amino acid metabolism pathway [98]. Similarly, the researchers used LC-MS to identify the effects of saponification and non-saponification astaxanthin on the metabolites in the plasma of oxidative stress rats, and a total of 80 potential biomarkers were identified [99]. Further comparative analysis showed that saponification astaxanthin group had a better therapeutic effect on oxidative stress, and this therapeutic effect was also related to the changes of intestinal microorganisms [99]. In addition to analyzing the metabolic mechanism of these bioactive components in animals, metabolomics can also be combined with molecular docking technology, which will play a certain guiding role in the development of functional food in the future.

Metabolomics analysis of biological fluids predict novel biomarkers or specific biomarkers, which provides insights for the prevention and treatment of diseases. However, there is still a long way to go before this stage, and long-term clinical studies are needed to determine their specific effects in humans.

### 3.4. Application of Metabolomics in the Screening of Bioactives for Human Trials

To evaluate the safety and effectiveness of a new food or drug, clinical studies are an essential step [120]. Metabolomics is being used in clinical trials more and more [121]. The screening of bioactive components and the analysis of metabolic changes caused by diseases or bioactive components are important applications of metabolomics in clinical research.

Food intake biomarkers (FIBs) are specific metabolic components produced after the consumption of a specific food. The existing dose and time have a clear response range which are important indicators for assessing their dietary intake in humans [122]. Non-targeted metabolomics was used to search for possible FIBs of several foods [100]. It was reported that galactose and galactonate levels in the serum of 11 healthy volunteers were significantly increased after drinking milk for 1 h, using GC-MS and ^1^H NMR, lactic acid levels in 1 h and 2 h after eating cheese were significantly higher than that in milk and soybean drinks, and Pinitol levels significantly increased after intaking of soy beverages [100]. Therefore, these altered metabolites were considered as candidate FIBs for dairy products and alternative dairy products. The results were also validated in the urine samples [100]. Although there were shortcomings, this study provided a new direction for searching FIBs in food. 

Drinking alcohol in moderation is thought to be beneficial for health because of the presence of phenols in alcohol [123]. Adelaida esteban-fernandez et al. investigated the effects of moderate red wine intake on human metabolism through non-targeted metabolomics, and the results showed that the phenol metabolites from microorganisms such as 5-(dihydroxyphenyl)-γ-valerolactones and 4-hydroxyl-5-(phenyl)-valeric acids were significantly up-regulated, indicating that the intake of red wine had a certain impact on the intestinal microbial balance [101]. Urine metabolome and fecal metabolome confirmed the conclusions [101]. The effects of garlic supplements on human metabolism can also be studied using metabolomics. The researchers used HPLC-ESI-QTOF-MS to identify metabolites in the plasma of volunteers after garlic supplements treatment, and found that 26 metabolites were significantly changed [102]. Further metabolomics studies showed that garlic supplements could improve human immune function mainly through phospholipid metabolism [102]. Moreover, garlic contains anti-glycosylation components, which has potential therapeutic function for diabetes and other chronic diseases [102]. Another study used targeted metabolomics to analyze the metabolites in blood and urine of adult patients with prediabetes after drinking a (poly) phenols-rich test drink for 24 h, and quantified a total of 110 gut microbial derived (poly) phenolic metabolites using UHPLC-QQQ [103]. The results showed that the levels of 3,8-dihydroxy-urolithin derivatives in plasma and phenyl-γ-valerolactones in urine were significantly reduced, consistent with the changes of some specific intestinal microbes, and it indicated that changes in intestinal microbial population of diabetic patients resulted in changes in the metabolism of β-cationic substances [103]. It provided a new reference direction and therapeutic target for the treatment of chronic metabolic diseases such as diabetes. Phanit Songvut and colleagues analyzed plasma metabolites in volunteers after taking extract of *Centella asiatica* (ECa 233) using ^1^H NMR and LC-MS/MS and found that the levels of choline in the plasma of the volunteers was significantly increased after taking ECa 233 500 mg/day [104]. The results indicated that metabolomics also has some contribution to the determination of drug bioavailability. Although the application of metabolomics in humans is relatively small, the above content is sufficient to reflect the powerful role of metabolomics in identifying the effective ingredients of food. Tt will make important contributions to the supplement of human nutrition and the regulation of chronic diseases. 

Application of metabolomics in clinical trials could provide a comprehensive understanding of the overall situation in humans. It can be used not only to discover the changes and effects of different bioactive substances on human metabolites, but also to determine the health effects of various bioactive ingredients. However, the changes of metabolites may be affected to some extent by physical conditions, health status, and living habits of the individuals. The long-term accumulation of unknown components, the measurement deviation caused by the rapid digestion and absorption of some substances, the interference caused by different excretion ways of metabolites, and the limitation of the number of people will all bring a lot of inconvenience to clinical observation.

## 4. Conclusions

Metabolomics as a new omics technology is gradually being accepted by researchers and has made significant research progress so far. Compared with proteomics and genomics, metabolomics is more correlated with physiology. At present, this method has been applied in medicine, nutrition, toxicology, and other fields. The key of metabolomics analysis is to interpret the biological information underlying a large amount of raw data through processing and statistics. Metabolomics is mainly the study of endogenous and exogenous compounds or metabolites in living organisms. It can not only be used to evaluate the metabolism of nutrients in vivo and in vitro, but also to evaluate their influences on biochemical environment after diseases or treatments. It will help us to understand the pathogenesis of chronic diseases, predict biomarkers of related diseases, analyze the composition of food, and evaluate the quality and safety of food. Metabolomics can be used to analyze and identify abnormal metabolites to find biomarkers leading to certain diseases, which is helpful for disease diagnosis and new drug development. It can also be used to predict and analyze the metabolites changes after food or diet intake and extend the bioactive substances for the development of functional food or drugs. Nutritional metabolomics is the use of metabolomics to study the interaction between diet and metabolism under different health status or disease conditions. It will help us to discover and develop new drugs or functional foods and provide approaches and tools for mechanistic research.

## 5. Challenges and Future Perspectives

At present, metabolomics techniques are still not able to analyze all metabolites produced in metabolic pathways, as some metabolites levels are too low to be detected as bioactive substances, or distorted into false positives by combination effects [124]. Experimental data are greatly affected by non-experimental factors. The main methods used for data analysis are PLS and PCA, which are mainly applied to linear data and rarely applied to nonlinear data. However, some data are not linear, and the processing of non-linear data is complex [125]. Using the above methods may cause the loss of non-linear correlation between samples. The inapplicability of each technology platform, the incompleteness of database, and the triviality of each step need to be further optimized. The underlying mechanisms of how bioactives play their roles needs to be further investigated. Many factors such as age, gender, weight, lifestyle, health status, and other individual differences need to be further studied. There is still a lot to be improved, including the length of observation time, the effect of digestion and absorption on the bioactive components, the discovery of metabolic pathways, and the accuracy of qualitative and quantitative analysis of metabolites. In summary, metabolomics research in the clinical trials need to be strengthened to accelerate the development of functional foods and drugs.

With the continuous development of metabolomics, it has made more and more contributions to the scientific field as a tool for analysis and prediction. However, in order to gain more recognition, more sensitive, fast, and advanced analysis and processing tools will become an indispensable requirement for the development of metabolomics in the future. Due to the immaturity and incompleteness of analytical instruments, techniques, and the operation of data collection, it is necessary to improve and establish a rational system for clinical diagnosis. For example, combining multiple omics platforms, increasing the number of metabolites recognized, optimizing each processing step, shortening the experimental time, and improving the sensitivity of the instrument should all be improved to increase repeatability. As an important analytical tool of metabolomics, the application of database is also indispensable. For example, HMDB is the most important human metabolome database, including metabolites, NMR data, MS data, and spectral data, and has been applied in many fields such as lipidomics, biochemistry, nutrition science, and so on [126]. METLIN molecular standards database is a biochemical database with more than 850,000 molecular standards [127]. The expanded and supplemented database METLIN MS^2^ can not only clarify the molecular structure, but also identify more microbial and human metabolites [127]. For the search of information on the structure and function of volatile organic compounds, KNApSAcK Metabolite Ecology Database is a good choice [128]. In addition, a comprehensive R package, MetaboAnalystR, has been developed recently, involving 11 analysis modules and including more than 500 functions, which can be used to analyze compounds, pathways, metabolites, and other related information. [129]. A cloud-based knowledge database called “The Metabolome of Food” is currently being built, and it includes human and microbial metabolites, phytochemical composition, and metabolic pathways [130]. In the future, it may become one of the important databases in the field of food nutrition. At present, various databases have been updated. From 2007 to 2022, for example, the number of compounds in the HMDB increased from 2180 to 217,920 [126,131]. The databases need to be continuously supplemented and updated with more new biomarkers. For the application of metabolomics in food-derived bioactive ingredients, based on in vitro and in vivo experiments, more studies on human trials are still needed to understand the health effects of bioactive ingredients in humans. It is also necessary to expand bioactive components and improve their bioavailability. Metabolomics is increasingly used in the field of food and nutrition. It will become an important technology for the development of functional food in the future.

## Figures and Tables

**Figure 1 foods-11-02974-f001:**
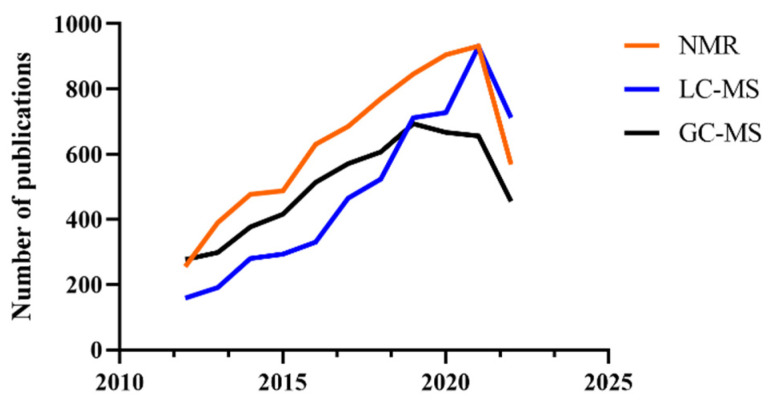
Numbers of publications using NMR, LC-MS, and GC-MS over the years for metabolomics analysis in the last decade by searching PubMed database with the keywords of metabolomics and NMR/LC-MS/GC-MS.

**Figure 2 foods-11-02974-f002:**
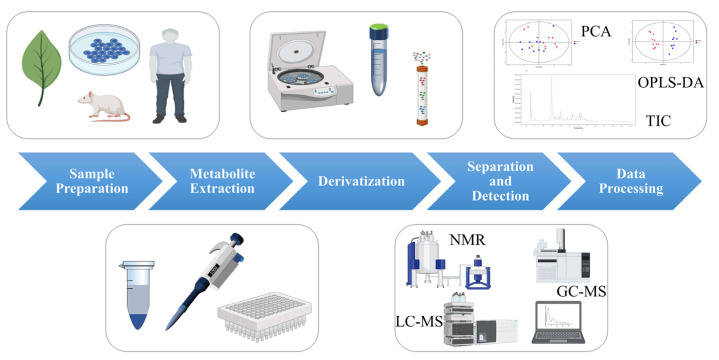
The process of metabolomics analysis. (Some picture elements are from the BioRender).

**Table 1 foods-11-02974-t001:** Metabolomics applications in food-derived bioactive ingredients.

Main Metabolites	Sample (Sources)	Analytical Technique	Application of Metabolomics	Reference
Flavonoids	Soybean seeds	LC-ESI-MS/MS	Evaluated the dynamic changes of metabolites in soybean seeds before and after germination.	[77]
Amino acids, sugars, choline	Mung bean	NMR	Evaluated the dynamic changes of metabolites in mung bean at different germination stages.	[78]
Flavonoids and polyphenols	Green tea bud	UPLC-QTOF-MS	Combined the characteristic metabolites with in vitro biological activities to determine the health effects of natural metabolites.	[79]
Isoflavones and alkaloids	*Lupinus albus* fractions	^1^H NMR UHPLC-ESI-MS/MS	Identified the effects of different extract components on the bioactivities of metabolites.	[80]
Procyanidin C1, orientin, quercetin, etc.	Hawthorn	UHPLC-Q-TOF/MS LC-MS/MS	Screened the metabolites with specific biological activities by combining various types of stoichiometry.	[81]
Polyphenols, glucosinolates and monomeric anthocyanins	Four Brassicaceae microgreens	UHPLC-QTOF	Compared the changes of metabolite concentrations before and after simulated gastrointestinal digestion in vitro.	[82]
Polyphenol	Red beet and amaranth	UHPLC-QTOF	Identified the effects of different storage periods on metabolite changes.	[83]
Phenyllactate and ferulate	Soybean protein hydrolysate	UHPLC/MS/MS_2_ GC/MS	Analyzed the compounds with significant effects on cell growth and IgG production.	[84]
Ornithine and citrulline	Soybean hydrolysates	LC-MS/MS	Screened productivity markers by comparing cell growth condition.	[85]
Phenolic substances	Red kidney bean extracts	NMR LC-MS	Analyzed the antiproliferative mechanism of different chemical components on B16-F10 melanoma cells.	[86]
Alanine, aspartate and glutamate	CG	UPLC-MS/MS	Studied the effects of different concentrations of CG on L-02 cells metabolism.	[87]
Glutamate and lactate	Exendin-4	NMR	Investigated the mechanism of protective effect of exendin-4 on mouse glomerulus mesangial cells.	[88]
Glycerolipid, cyanomino acid, inositol phosphate, etc.	Vitamin C	^1^H NMR	Determined the effect of half inhibitory concentration of Vitamin C on cell metabolism.	[89]
Alanine, Aspartate, glutamate, etc.	Emodin	^1^H NMR	Evaluated the cytotoxic effects of high concentrations of emodin on cells.	[90]
Lactate and glucose	Doxorubicin and dexrazoxane	^1^H NMR	Identified the important factor of dextroprazole induced cardiotoxicity.	[91]
Nicotinamide, nicotinic acid, Arginine, etc.	Quinoa saponins	UHPLC-MS	Combined the metabolomics with the changes of intestinal microbes in rats and identified the differential effects of quinoa saponins on different sexes.	[92]
Phosphatidylcholine and palmitic acid	Corn silk	UPLC-ESI-Q-TOF/MS	Identified the changes of diabetes markers through the differences of serum metabolites in rats.	[93]
Flavonoids	Fenugreek	UPLC-Q-TOF-MS	Investigated the function of fenugreek flavonoids in regulating blood glucose by serum metabolomics.	[94]
Valine, leucine, LPCs, etc.	RS3	UHPLC-LTQ/Orbitrap MS	Identified the antidiabetic mechanism of RS3 by urine metabolomics.	[95]
Alanine, aspartate, glutamate, etc.	GAP	LC-MS	Studied the regulation of GAP on mice with nonalcoholic fatty liver by serum metabolomics.	[96]
Phenylalanine, tyrosine and tryptophan	EP	LC-MS	Combined the metabolomics with molecular docking technology to obtain effective bioactive components.	[97]
Arginine and proline	The hydrolysates of yak bone glue	UPLC-QTF/MS	Determined the anti-obesity mechanism of the hydrolysates of yak bone glue by fecal metabolomics.	[98]
Propionic acid, taurine, glutathione, etc.	Astaxanthin	LC-MS	Clarified the mechanism of astaxanthin alleviating oxidative stress in rats.	[99]
Galactose, galactonate and lactic	Cheese, milk and soy beverages	GC-MS ^1^H NMR	Explored possible food biomarkers of human intake by metabolomics.	[100]
5-(dihydroxyphenyl)-γ-valerolactones and 4-hydroxyl-5-(phenyl)-valeric acids	Red wine	UHPLC−TOF-MS	Determined the health effects of moderate red wine consumption on human metabolism by urine metabolomics and fecal metabolomics.	[101]
Lysophosphatidylcholines, lysophosphatidylethanolamines and acylcarnitines	Garlic supplements	HPLC-ESI-QTOF-MS	Verified the function of garlic supplement in enhancing immunity by fingerprint metabolomics.	[102]
3,8-dihydroxy-urolithin derivatives and phenyl-γ-valerolactones	A (poly) phenols-rich test drink	UHPLC-QQQ	Determined the regulating mechanism of polyphenol beverage on diabetes patients by blood and urine metabolomics.	[103]
Choline	ECa 233	^1^H NMR LC-MS/MS	Evaluated the drug bioavailability of ECa 233 by metabolomics.	[104]

## Data Availability

No data were provided in the study.

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
