# Peer review of "Innovative Application of Metabolomics on Bioactive Ingredients of Foods"

_foods, 2022, doi:10.3390/foods11192974_

Round 1
Reviewer 1 Report
The authors focused on a popular technology, metabolomics, which has been widely applied in the field of food sciences. And this review specified on the innovation of practical applications. Indeed, this is a hot topic in the field, at the same time, highly creative and inspired viewpoints are required.
Page 1, Line 43: Can the authors summarize the overall features and developing trends for all the three approaches? Besides, is there any evidence of their relative popularity (e.g., making a line chart)?
Page 2, Line 51: Compared to 1H-NMR, heteronuclear NMR (e.g., 31P, 15N) seem more uncommon and thus, potential to lead to exquisite works. Therefore, the authors are suggested to describe more on heteronuclear NMR-based applications, or opinions of this approach.
Page 2, Line 60: This paragraph seems strange here. May it be moved to somewhere else?
Page 3, Line 85: Sample preparation is actually closely connected to detection platform. I strongly suggest the authors concern detection together with pretreatment, and clearly indicate the innovative points happening during which approach.
Page 3, Line 117: there are not a little language and format issues. For example, here, the “subscript” issues in the chemical names. Besides, the style (full name or abbreviation) should be unified.
Page 3, Line 140: Please focus more on the “innovative points”, such as the new strategy of detection, optimization of parameters, special instruments (or parts), customized manipulators, etc.
Page 4, Line 191: PCA and DA should be the most classical and popular statistical methods in this field. Are there any new thoughts of dealing with such data? Please the authors give their ideas of new statistical thoughts.
Page 5, Line 220: I would like to see a summarized table listing all the reviewed applications and highlighting their innovative points.
Page 11, Line 521: Databases are indeed important tools. However, are the databases referring to MS-based platforms only (or NMR only)? For example, HMDB is a combination of metabolism, structure, MS data, NMR data… And there are other more commercial or non-commercial libraries. How about giving a collective description? That will be helpful for the authors who would like to learn something from this review.
Reviewer 2 Report
Manuscript send to review with title: ” Innovative Application of Metabolomics on Nutritional Compounds in Foods” influenced by different rootstocks” is very interesting and well written.
Presented manuscript is on good scientific level and represent a Review manuscript type.
Abstract. Authors give a short presentation of manuscript.
Introduction section.
The Introduction section includes all necessary information about topic. For better understanding of manuscript Authors share the whole manuscript for sub-sections. In my opinion each sections are described with all details. The order of appearance of the sections and their description is very logical and correct. I am impressed with the work done by the authors of the manuscript.
Conclusions. Authors give a very good conclusions which correspond with manuscript text. More over Challenge and future perspectives section are important as well.
General opinion: After carefully manuscript reading, I think, that presented manuscript is a very valuable.
Reviewer 3 Report
The manuscript presents an overview of metabolomics and its application for health and nutrition. The text is clear, but it might be difficult for those who are not familiar with the techniques mentioned in the manuscript such as MNR and LC-MS. I also missed figures (MNR spectra, PCA, MS spectra) to illustrate the metabolomics results. Those tools are clear for readers who works with them, the general audience will find the manuscript hard to understand without illustrations.
Thus, I strongly suggest to insert figures allowing the reader to visualize the analytical techniques and metabolomic results
Round 2
Reviewer 1 Report
The authors have provided more descriptions and added necessary tables/figures according to the reviewer’s comments. I believe that the current version is much more comprehensive and better organized. So, in my opinion, it should be accepted.
Reviewer 3 Report
The suggested changes and improvements were done in the manuscript text.